# Community Pharmacists’ Knowledge, Attitudes and the Perceived Safety and Effectiveness of Melatonin Supplements: A Cross-Sectional Survey

**DOI:** 10.3390/pharmacy11050147

**Published:** 2023-09-15

**Authors:** Mansour Tobaiqy, Faris A. AlZahrani, Abdulrahman S. Hassan, Abdullah H. Alirbidi, Osama A. Alraddadi, Omar A. AlSadah, Mohammad B. Yamani, Sulafa T. Alqutub

**Affiliations:** 1Department of Pharmacology, College of Medicine, University of Jeddah, Jeddah P.O. Box 45311, Saudi Arabia; 2College of Medicine, University of Jeddah, Jeddah P.O. Box 45311, Saudi Arabia; 2040418@uj.edu.sa (F.A.A.); 2040067@uj.edu.sa (M.B.Y.); 3Department of Family and Community Medicine, College of Medicine, University of Jeddah, Jeddah P.O. Box 45311, Saudi Arabia; stalqutub@uj.edu.sa

**Keywords:** melatonin supplements, community pharmacists, knowledge, attitudes, safety, effectiveness

## Abstract

Melatonin, which is classified as a dietary supplement by the Saudi Food and Drug Authority, is used to manage sleep disorders. In this study, community pharmacists’ knowledge and attitudes about dispensing melatonin supplements and the perceived safety and effectiveness of melatonin were assessed. A cross-sectional survey of community pharmacists in Jeddah, Saudi Arabia was conducted from March–June 2023. Community pharmacists’ knowledge and attitudes towards prescribing and dispensing melatonin supplements, the methods of dispensing melatonin supplements (prescription, over the counter, self-administered), indications, ages of users, dosage forms, and adverse drug reactions related to melatonin use among consumers were surveyed using a questionnaire. Potential participants were approached face to face, a questionnaire was administered to those agreeing to participate in the study, and responses were recorded electronically. The response rate of the 300 community pharmacists who participated in this study was 83.5%. The mean age of participants was 33.6 years, and 89.3% (*n* = 268) of community pharmacists reported dispensing melatonin supplements. Self-medication and over the counter (*n* = 213; 71.2%) were the most common dispensing and prescribing patterns (*p* = 0.001). Awareness rates about melatonin supplement pharmacokinetics and pharmacodynamics among community pharmacists were 38% and 37%, respectively. Despite its popularity, community pharmacists reported relatively low rates of awareness of melatonin supplement pharmacokinetics and pharmacodynamics. Further attention to this issue is needed.

## 1. Introduction

Community pharmacists in Saudi Arabia (S.A.) are among the first-line healthcare providers that can recommend non-prescribed over-the-counter (OTC) medications and educate and advise patients and their families [1]. In S.A., the demand for quality community pharmacists’ services has increased due to a shortage of trained primary healthcare physicians and the increased number of patients with chronic diseases [2,3].

One of the most commonly prescribed and dispensed supplements in the community is melatonin [4,5]. Melatonin is a neurohormone naturally produced by the pineal gland in the brain. Melatonin regulates the circadian rhythm, which governs the sleep–wake cycle [1]. Melatonin also modulates the circadian rhythm by binding to two types of melatonin receptors: MT1 and MT2. MT1 receptors are primarily expressed in the brain. In contrast, MT2 receptors are expressed in the brain, gastrointestinal tract, liver, kidney, and all biological fluids, including cerebrospinal fluid, saliva, bile, amniotic fluid, and breast milk [6,7]. Sleep disorders, such as insomnia, obstructive sleep apnea (OSA), and restless leg syndrome (RLS), are increasing worldwide and within any community with a sedentary lifestyle [8,9]. Melatonin has been found to improve sleep quality in sleep disorders, including insomnia, OSA, and RLS [9,10]. A systematic review and meta-analysis study was conducted to determine the effectiveness of melatonin in improving sleep. The researchers identified 5030 studies and selected 12 based on their inclusion criteria: double- or single-blind, randomized and controlled. The results showed that melatonin use was most effective in reducing sleep onset latency in primary insomnia (*p* = 0.002), delayed sleep phase syndrome (*p* < 0.0001), and regulating the sleep–wake patterns in blind arm patients compared with placebo. These findings suggest that melatonin may effectively treat certain first-degree sleep disorders. However, further evidence is needed from large-scale, randomized, controlled trials to support its therapeutic use in various sleep difficulties [11].

The prevalence of OSA risk and symptoms in middle-aged Saudi adults is 3 in 10 among men and 4 in 10 among women [12]. Among male and female university students, the rates of insomnia and circadian rhythm sleep disorders are 33% and 22.4%, respectively [13,14].

Melatonin is considered a dietary supplement (D.S.) by the U.S. Food and Drug Administration (FDA). Melatonin is approved and used as an OTC medicine in Europe, Australia, and Japan and can be obtained without a prescription [15]. Melatonin is also listed as a D.S. by the Saudi Food and Drug Authority (SFDA) [16].

The increasing use of melatonin co-administered with other drugs raises the possibility of interactions leading to elevated melatonin levels. Melatonin is extensively metabolised primarily through hydroxylation at the 6-position, catalysed selectively by the microsomal CYP1A2 enzyme of the cytochrome P450 superfamily. It undergoes rapid first-pass metabolism and has a short half-life of approximately 30–60 min. Metabolites are mainly excreted in the urine [17,18,19].

Following intravenous administration, melatonin reaches peak plasma levels in approximately 0.5 to 0.6 min, whereas oral administration results in peak plasma concentration in approximately 60 min [17,18,19].

A systematic review investigating the pharmacokinetics of exogenous melatonin in humans found that, despite methodological differences between the included studies, the time to maximal plasma concentration (Tmax) was approximately 50 min following oral immediate-release melatonin formulations. The elimination half-life (T1/2) was 45 min for both oral and intravenous administration routes. The maximal plasma concentration (Cmax), area-under-the-curve plasma concentrations (AUC), clearance (Cl), and volume of distribution (VD) varied extensively between studies. The bioavailability of oral melatonin was approximately 15% [17].

The plasma levels of melatonin may be influenced as a result of concurrent exposure to chemicals, including drugs that modulate the expression of CYP1A2; for example, plasma melatonin levels were increased following fluvoxamine administration, presumably by impairing its cytochrome P450-mediated metabolism; fluvoxamine is a potent inhibitor of CYP1A2. Melatonin’s pharmacokinetics may also be affected by age, caffeine intake, smoking and oral contraceptives [17,18,19].

Adverse drug reactions (ADRs) are a major health issue which increases drug-related morbidity and mortality. ADRs have been defined by the World Health Organization (WHO) as “a response to a drug that is noxious and unintended and which occurs at doses normally used for the prophylaxis, diagnosis or therapy of disease, or for modification of physiological function” (World Health Organization, 1966) [20].

Pharmacovigilance is the process of detection, assessment, understanding and prevention of short- and long-term ADRs in adult and paediatric populations, and plays a key role in patient medication safety [20,21].

Animal and human studies have shown that short-term use of melatonin is generally safe, even in high doses. Only mild adverse effects have been reported, such as dizziness, headache, nausea, and sleepiness. No studies have found that exogenous melatonin causes severe adverse effects [22,23].

Reported adverse events associated with long-term melatonin use are low, and few clinically significant adverse events have been reported. However, the scarcity of data from double-blind, randomised, placebo-controlled trials necessitate the analysis of large databases to provide high-quality evidence on which to base a more rigorous evaluation of the safety profile of melatonin [22,23].

Furthermore, melatonin supplement consumption has significantly increased in recent years among United States adults from 0.4% in 1999–2000 to 2.1% in 2017–2018 and similarly has risen from 2.0 to 19.9 per 1000 people between 2008 and 2019 in England [4,5].

Although paediatric melatonin dosing can be challenging, manufacturers in the United States have begun producing melatonin products specifically targeted at children [24]. A double-blind, placebo-controlled study of 40 children aged 6–12 years with more than one year of chronic sleep onset insomnia found that 5 mg of melatonin taken at 6 PM was relatively safe to administer in the short term and significantly more effective than placebo in advancing sleep onset and increasing sleep duration [25].

In Saudi Arabia, melatonin supplements are not licensed and different formulations and dosage forms are available without restrictions in community pharmacies [1,16]. In contrast, the first UK-licensed melatonin preparation was available in 2007 for the short-term treatment of primary insomnia in adults over 55 years, and recently, it has been licensed for treating insomnia in children with autism or Smith–Magenis Syndrome [5].

Over-the-counter medicine misuse is a recognised international issue, and the increasing availability of these medicines and dietary supplements might pose a risk for potential adverse events [26]. A study indicated that U.S. sales of melatonin increased by about 150% between 2016 and 2020. During 2012–2021, the annual number of paediatric ingestions of melatonin increased by 530%, with a total of 260,435 ingestions reported. Furthermore, paediatric hospitalisations and more severe outcomes similarly increased due to unintentional melatonin ingestions in children aged ≤5 years; gastrointestinal, cardiovascular, and central nervous systems were the most reported adverse effects [27].

In 2022, AlHazmi et al. reported regional variations in community pharmacists’ practice quality within S.A. and suggested that targeted professional development can overcome these gaps in practice [1]. Inappropriate use and misuse of OTC medications and D.S. have been reported in S.A. by both community pharmacists and the general public; however, information on the misuse of melatonin supplements has yet to be reported [28,29].

In Saudi Arabia, pharmacovigilance is considered a new concept, where the SFDA is the organising body for all pharmacovigilance activities in the country [30]. Health institutions, marketing authorisation holders and healthcare professionals are involved in pharmacovigilance activities and, more recently, during the COVID-19 vaccination program, direct patient reporting was established [31].

Recently, the SFDA has warned against the excessive use of melatonin supplements. The authority highlighted that excessive melatonin supplement use may cause headaches, nausea and dizziness [32].

Moreover, international health systems should provide integrated and collaborative models to improve the management of general sleep health, insomnia, and OSA in pharmacy practices [33]. Hence, this study aimed to assess community pharmacists’ knowledge and attitudes toward dispensing melatonin supplements and their perceived safety and effectiveness.

## 2. Materials and Methods

### 2.1. Study Design and Sample Size

The study was designed as a cross-sectional survey of licensed community pharmacists employed in the private sector in Jeddah, S.A. from March–June 2023. According to a recent report published by the Saudi Arabian Ministry of Health, 3228 community pharmacists work in the private sector in Jeddah [34]. Based on the previous estimates of 50% response distribution rates [26], the following formula was used to calculate the sample size in Open Epi [35]:*n* = [DEFF × Np(1 − p)]/[(d^2^/Z^2^_1−α/2_ × (N − 1) + p × (1 − p)]gt6

The above equation used a 95% confidence level and a 5% margin of error. The calculated sample size was 344. However, to account for potential loss to follow-up, 359 community pharmacists were invited to participate in the study.

### 2.2. Questionnaire Development

A 20-item questionnaire with both a Likert scale and open-ended questions was produced based on previous questionnaires [36,37]. The questionnaire was anonymous and included questions about demographics (sex, age, undergraduate pharmacy education, qualifications, and work experience), community pharmacists’ knowledge and attitudes toward prescribing and dispensing melatonin supplements, how users obtained melatonin supplements (prescription or OTC self-administered), indications, age of users, dosages, and forms. Questions about the community pharmacists’ awareness of the pharmacokinetics and pharmacodynamics of melatonin and the community pharmacists’ attitudes concerning the safety, effectiveness, and abuse of melatonin were also included in the questionnaire. In addition, adverse drug reactions related to melatonin use that were witnessed by the community pharmacists during the last three months was recorded. Before use, the questionnaire was reviewed for face and content validity by a panel of ten individuals representing academic, healthcare, and administrative staff from the medical centre at the University of Jeddah. The questionnaire was then tested by ten community pharmacists. The pilot responses were included in the analysis as no changes were made to the questionnaire after the pilot test.

### 2.3. Recruitment

A random sample of pharmacists who work in different independent and chain community pharmacies (*n* = 359) were invited to participate in the study. Face-to-face administration of the survey was conducted from March to June 2023. This approach was thought to increase engagement and be more personal and convenient for the community pharmacists. Each potential participant was informed about the study objectives then had the opportunity to ask any questions about the study. Each community pharmacist who agreed to participate in the study was asked to give their informed consent before the survey was administered face to face. The research team members electronically recorded the responses.

### 2.4. Analysis

Descriptive statistics with frequency tables were generated for the sociodemographic variables and variables related to melatonin prescription forms and patterns. Chi-square or Fisher’s Exact tests were applied to test for significant associations between the awareness of melatonin pharmacokinetics and pharmacodynamics, and the dispensing and prescribing (pharmacists’ recommendation) of melatonin supplements by the community pharmacists. The independent variables were as follows: community pharmacists’ sex, qualifications, faculty classification, years of experience, observed adverse drug reactions reported by melatonin supplement users, and the perceived safety and effectiveness of melatonin supplements used for sleep disorders. A *p*-value < 0.05 was considered significant. Data analysis was performed using SPSS (SPSS version 22.0).

## 3. Results

### 3.1. Community Pharmacists’ Demographics

Of the 359 community pharmacists (CPs) approached for this study, 300 CPs agreed to participate in the study, giving a response rate of 83.5% (300/359). The mean age of the participants was 33.6 years (standard deviation, 6.9) (Table 1). More male CPs reported dispensing and prescribing melatonin supplements (*n* = 231; 86.2%) compared to female community pharmacists; however, the sex difference was not statistically significant. Insomnia was the most frequent (60% of the responses) indication for melatonin supplements. The following factors were not significantly associated with differences in the pattern of dispensing or prescribing melatonin supplements: CP qualifications, faculty classification, and years of pharmacy work experience.

### 3.2. Estimated p-Value Chi Square Test Applied

The most frequent dispensing and prescribing pattern for melatonin supplements was self-medication, over the counter capsules, with a 3 mg strength (*p*-value = 0.001) (Table 2).

### 3.3. Knowledge/Experience of the CPs (Community Pharmacists)

Eighty-four percent of community pharmacists dispensed or prescribed melatonin without being aware of its pharmacokinetics (*n* = 156; 84.3%) (Table 3), and pharmacokinetic unawareness significantly affected the frequency of dispensing and prescribing melatonin (*p*-value < 0.001) (Table 4). In addition, 84.7% (*n* = 161) of community pharmacists who dispensed or prescribed melatonin were unaware of the pharmacodynamics of melatonin supplements, and pharmacodynamic unawareness significantly affected the frequency of dispensing and prescribing melatonin (*p*-value < 0.001) (Table 4). Drug adverse reactions were the highest among male users, but this did not significantly affect the frequency of dispensing or prescribing melatonin (Table 3 and Table 4).

### 3.4. CPs’ Attitude towards Melatonin Safety and Effectiveness and for Sleep Disorder

Community pharmacists who reported dispensing or recommending melatonin either agreed or strongly agreed that asking users about their past medical history before prescribing or dispensing melatonin was important (*n* = 164; 61.2%); the difference in the frequency of dispensing and prescribing melatonin was statistically significant (Chi-square test, *p*-value < 0.001). Community pharmacists who reported dispensing and prescribing melatonin either agreed or strongly agreed that melatonin supplements are effective in treating sleep disorders among adults (*n* = 248; 92.5%); the effects on the frequency of dispensing or prescribing melatonin were statistically significant (Fisher’s Exact test, *p*-value < 0.02). A relatively large proportion of community pharmacists perceived the effectiveness of melatonin supplements in treating sleep disorders in pediatric patients and the general safety of melatonin; however, this awareness did not significantly affect the frequency of dispensing or prescribing melatonin by the community pharmacists (Table 5).

## 4. Discussion

The awareness rates among community pharmacists in Jeddah about the pharmacokinetics and pharmacodynamics of melatonin supplements was only 38% and 37%, respectively (Table 3). Several studies identified challenges and issues concerning the knowledge of pharmacology and clinical pharmacology among community pharmacists [38,39] and junior doctors [40] that need substantial improvement and training. Community pharmacists reported a high rate of melatonin supplement dispensing in a sample of community pharmacies in Jeddah: 89%. A significant inverse relationship was observed between the knowledge about melatonin pharmacokinetics and pharmacodynamics and dispensing rates. Nevertheless, a recent Saudi study reported a 20% prevalence of melatonin supplement use, which ranked second after vitamin D as a D.S. to support mental health [41]. The higher-than-expected dispensing and prescribing pattern for melatonin supplements may be due to several reasons. First, community pharmacists cannot influence or control melatonin-dispensing patterns of OTC supplements [16]. Second, melatonin consumption increased during the post-COVID-19 era as it was recommended as an adjuvant for SARS-Cov-2 management [42,43]. Third, despite the questionnaire validation, cognitive recall bias is weak in cross-sectional surveys [44].

A significant positive relationship was found between the perceived effectiveness of melatonin supplements in treating sleep disorders and dispensing patterns among adults. This finding is consistent with the available evidence on the effectiveness of melatonin supplements in treating insomnia and regulating the sleep–wake cycle [9,41]. However, no significant relationship was found between the perceived effectiveness and the dispensing pattern in the paediatric age group. Further studies are needed to uncover patterns and indications of melatonin supplement use in the paediatric age group, especially for children older than 12 [45].

A recent systematic review aimed at developing a recommendation on melatonin use for children and adolescents aged 5–20 years with chronic insomnia revealed that melatonin moderately improved sleep continuity measures, such as sleep duration and onset, but did not affect sleep quality and daytime functioning. The mean increase in sleep duration was 30.33 min, and the mean decrease in sleep onset was 18.03 min [46]. However, the authors rated the certainty of evidence as very low for sleep quality and low for daytime functioning based on the risk of bias, inconsistency, and imprecision of the included studies [46].

It is essential to note that most studies evaluating melatonin use in the pediatric population have been conducted with children with comorbidities, such as autism spectrum disorder or attention-deficit/hyperactivity disorder (ADHD) [11].

Community pharmacists in this study have reported dispensing melatonin supplements in a maximum daily dose of 3 mg (31%), followed by 5 mg (26.3%) and 6 mg (24.7%), respectively. According to a clinical review, doses of melatonin between 1 mg and 6 mg appear to be effective in improving sleep disorders in older adults [47,48].

Another study suggested that doses between 0.5 and 5 mg had comparable efficacy, but higher melatonin doses may induce more sedation. In the clinical trials, the daily melatonin doses ranged from 0.15 mg to 12 mg, and a dose of 20 mg did not alleviate fatigue or other symptoms in patients with advanced cancer [49,50].

Despite the relatively high rates of perceived safety, no significant difference in dispensing patterns was detected. Approximately 25% of community pharmacists perceived that consumers misuse melatonin supplements. In S.A., a qualitative study about community pharmacists’ views and experience with OTC medication misuse demonstrated the challenges encountered by practicing pharmacists, especially with commonly misused sedating antihistamines and analgesics. However, misuse of melatonin was overlooked and not reported in that study [30].

The global use of medicines and the occurrence of self-medication have been increased by the COVID-19 pandemic, particularly during its initial stages when no approved treatments or vaccines were available for managing the disease [4,5,42]. While a quarter of the participants in this study believed that melatonin supplements had been misused, it is essential to note that pharmacists play a crucial role in preventing and reducing drug misuse and should be involved in evidence-based actions to detect, understand, and prevent drug diversion activities and the adverse effects of drug misuse. Community pharmacists can aid in preventing the misuse of OTC drugs, including melatonin, by educating and counselling patients on the proper use of these supplements. Community pharmacists can also monitor for potential drug interactions and advise patients on using OTC drugs in combination with other medications [51,52,53].

Additionally, pharmacists can work with other healthcare professionals to identify and address cases of OTC drug misuse. While melatonin is available as an OTC drug in some countries, it is classified as a prescription-only medication in others. Community pharmacists can play an essential role in ensuring its safe use by providing information on appropriate dosing and potential adverse effects and also advising patients on the proper use of melatonin for sleep-related disorders [51,52,53].

Only 8% of community pharmacists in this study observed adverse drug reactions associated with melatonin use. A recent systematic review that included 37 randomized clinical trials found that the most frequently reported adverse events were daytime sleepiness (1.66%), headache (0.74%), other sleep-related adverse events (0.74%), dizziness (0.74%), and hypothermia (0.62%). The study reported very few adverse events considered to be severe or of clinical significance, such as agitation, fatigue, mood swings, nightmares, skin irritation, and palpitations. Most adverse events either resolved spontaneously within a few days with no melatonin adjustment or immediately upon treatment withdrawal. Melatonin was generally regarded as safe and well-tolerated. However, there are insufficient robust data for a meaningful appraisal of concerns that melatonin may result in more clinically significant adverse effects in potentially at-risk populations.

Community pharmacists should be aware of the drawbacks of OTC medications, including misdiagnosis, misuse, overdose, and drug interactions [52]. Despite the safety of melatonin supplements, scheduling policies are recommended by the regulatory authority and potentially the SFDA. Regulation should ameliorate the chronobiotic characteristics of melatonin that increase the risk of misuse. 

## 5. Conclusions

Melatonin is commonly prescribed and dispensed to consumers as a self-administered over-the-counter supplement with considerable community pharmacist recommendations or physician prescriptions. This study revealed the need to improve community pharmacists’ knowledge concerning the pharmacokinetics, pharmacodynamics, and potential adverse drug reactions of melatonin. Community pharmacists should be aware of the drawbacks of OTC melatonin supplements, including misdiagnosis, misuse, overdoses and drug interactions. Practicing community pharmacists should advocate assessment by physicians before melatonin supplement use. In addition, the regulatory authority should apply scheduling policies for melatonin supplements to mitigate the risk of melatonin misuse. Although most survey participants believed that melatonin supplements are effective and safe, further research should focus on individuals using melatonin to identify safety issues and latent adverse outcomes.

### Limitations

Community pharmacists have provided valuable insights into the use of melatonin supplements. However, reporting consumer compliance rates, which would be a reliable indicator of the nature and use of these supplements, was not within the scope of this study. Additionally, the small sample size made it difficult to conduct regression analysis while controlling for confounding factors. It should also be noted that the study was limited to one location (Jeddah, Saudi Arabia) and was conducted over a short period (March–June 2023).

## Figures and Tables

**Table 1 pharmacy-11-00147-t001:** Characteristics of the participating community pharmacists and the reported melatonin supplement prescription patterns (*n* = 300).

Variable	Frequency (%)
Mean age (±SD)	33.63 (±6.9)
Sex	Male 254 (84.7%) Female 46 (15.3%)
Pharmacist qualifications	Master of Science 14 (4.7%) Pharm. D 73 (24.3%) Pharm BSc(Bachelor of Science) 197 (65.7%) PhD 16 (5.3%)
Undergraduate pharmacy faculty classification	S.A. governmental faculty 75 (25%) S.A. private faculty 22 (7.3%) Non-Saudi faculty 203 (67.7%)
Years of pharmacy work experience	1–5 years (30.3%) >5–10 106 (35.3%) >10–20 84 (28%) >20 19 (6.3%)
Reported dispensing and or prescribing melatonin by the participant	Yes 268 (89.3%) No 32 (10.7%)
Reported pattern of melatonin indications	Insomnia 178 (59.5%) Jet lag 58 (19.4%) Better sleep quality 56 (18.7%) Jet lag and insomnia 7 (2.3%)

**Table 2 pharmacy-11-00147-t002:** The dispensing patterns and common forms of melatonin supplements reported by community pharmacists (*n* = 300).

Variable	Frequency (%)	*p*-Value
Melatonin supplements most commonly purchased according to the following:	Physicians’ prescription 23 (7.7%) Pharmacists’ recommendation 63 (21.1%) Patient self-medication OTC 213 (71.2%)	0.001 *
Dosage forms of most commonly prescribed and dispensed oral melatonin supplements	Capsules 116 (46.6%) Film-coated tablets 72 (28.9%) Tablets 50 (20.1%) Gummies 7 (2.8%) Sublingual 4 (1.6%)	0.001 *
The most commonly dispended dose of melatonin supplments	1 mg 11 (3.7%) 3 mg 93 (31%) 5 mg 79 (26.3%) 6 mg 74 (24.7%) >6–15 mg 43 (14.3%)	0.001 *

* indicates significant difference for chi-square test.

**Table 3 pharmacy-11-00147-t003:** Knowledge and experience about pharmacokinetics and pharmacodynamics of melatonin supplements reported by community pharmacists (*n* = 300).

Question	Response
Are you aware of pharmacokinetic interactions and drug–drug interactions of melatonin supplements?	Yes 115 (38.3%) No 185 (61.7%)
Are you aware of pharmacodynamics and the adverse drug reactions of melatonin supplements?	Yes 110 (36.7%) No 190 (63.3%)
Have you observed any adverse drug reactions (ADRs) associated with the use of melatonin supplments in the last three months?	Yes 23 (7.7%) No 277 (92.3%)
In what patient group(s) do you recall the ADR occurring?	Males 44 (72.1%) Females 9 (14.8%) Pediatrics 3 (4.9%) Other group not specified 5 (8.2%)

**Table 4 pharmacy-11-00147-t004:** The association between unawareness to pharmacokinetics and pharmacodynamics to dispensing and prescribing melatonin.

Variable	Dispensing or Prescribing Melatonin	Not Dispensing Nor Prescribing Melatonin	*p*-Value
Reported unawareness on pharmacokinetics *n* = 185	156 (84.3%)	29 (15.7%)	<0.001 *
Reported unawareness of pharmacodynamics *n* = 190	161 (84.7%)	29 (15.3%)	<0.001 *
Reported adverse drug reactions over the last three months *n* = 23	23 (100%)	0	0.067

* Presents a significant difference in the chi-square test.

**Table 5 pharmacy-11-00147-t005:** Community pharmacists’ attitude toward safety and effectiveness of melatonin supplement use for sleep disorders (*n* = 300).

Item	Strongly Agree	Agree	Neutral	Disagree	Strongly Disagree
It is essential for me to take past medical history before prescribing or dispensing melatonin	115 (38.3%)	58 (19.3%)	72 (24%)	48 (16%)	7 (2.3%)
Melatonin is effective in adult sleep disorders	119 (39.7%)	156 (52%)	20 (6.7%)	3 (1%)	2 (0.7%)
Melatonin is effective in pediatric sleep disorders	58 (19.3%)	128 (42.7%)	54 (18%)	52 (17.3%)	8 (2.7%)
Melatonin is safe in general	140 (46.7%)	131 (43.7%)	20 (6.7%)	7 (2.3%)	2 (0.7%)
Melatonin supplements have been misused	18 (6.1%)	55 (18.6%)	89 (30.2%)	101 (34.2%)	32 (10.8%)

## Data Availability

The data presented in this study are available upon request from the corresponding author.

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
