# Peer review of "Community Pharmacists’ Knowledge, Attitudes and the Perceived Safety and Effectiveness of Melatonin Supplements: A Cross-Sectional Survey"

_pharmacy, 2023, doi:10.3390/pharmacy11050147_

Round 1

Reviewer 1 Report

Page 1 – line 28: clarify response rate. Wording is strange

Line 30 – 71.2% appears to refer to “self-medication” not use of 3mg capsules. Please clarify this statement in the abstract.

Line 55 – need to review the sources stated here, as well as 10 and 11. #10 is a poorly controlled trial of 12 elderly subjects. 11 is a review article, of limited relevance to the topic (there is only one small section discussing the evidence for melatonin in insomnia). Please cite more robust primary literature for the claims made here.

Line 80 – wording of “frequent possible Adverse effect” is unusual. Suggest rewording to reflect true meaning of statement.

Line 133 – Why is the calculated sample size different than the actual invited number to participate? Please clarify this item.

Line 136 – Would be more clear if it read “20-item questionnaire”

Line 137 – It is not clear why there is a citation to #32 and #33 here as the references seem to simply have used Google Forms in the past. Both of these citations appear to be the primary author’s own works. It is somewhat surprising that the Research Ethics Board would approve the use of Google Forms as survey software due to the lack of privacy and security of Google Forms, along with hosting the information outside of the researcher’s organization, although it may be reasonable if no personally identifying information was collected within the Google Form – can the authors attest to this?

Line 152 –  Suggest reorganizing this section as follows for clarity: “A sample of pharmacists who work in different independent and chain community 153 pharmacies (n=359) were invited to participate in the study.  Each potential participant was informed about the 155 study objectives, and community pharmacists who agreed to participate in the study was 156 asked to sign an informed consent. Interviews were conducted, and the research team 157 members electronically recorded the responses. Face-to-face interviews were 154 conducted from March to June 2023.”  Can the authors also clarify why this method was chosen, as opposed to simply sending the questionnaire to the pharmacists themselves?  The interview may have imported some bias into the responses as the interviewer would not be blinded to the responses. There is also the added potential for data input errors.

Line 165 – Conflating sex and gender, please correct.

Table 1 – define “BSC”

Table 1 – provide context to the sex distribution – is this proportional to the sex distribution of pharmacists in S.A.?

Table 2 – can the authors describe “patterns”. It appears based on the frequencies that each pharmacist could only select one response. Presumably it is possible for all of the options to be possible for each pharmacist. What was the question asked?  Most common/frequent? The same applies to other questions in Table 2

Table 3 – The two final questions have significant recall bias, and are of limited use.  Suggest rewording to indicate that this is the pharmacist’s recollection.  The wording of the final question is difficult to understand; suggest rewording to “In what patient group(s) do you recall the ADR occurring?” if appropriate.

Line 193 – differentiate between “dispensed” and “prescribed” earlier in the methods. In this case is “prescribed” referring to “Pharmacists’ recommendation” as it appears in Table 2? Ensure consistency of terminology throughout the document.

Line 193  and 195 – denominator being used is not the respondents (n=300)? – 112/300 = 37%  Please correct or clarify the denominator being used if not the total sample size.

Line 204 – “prescribing” referring to “recommending”?

Line 224 – This statement is not clearly linked to the presented results.  The authors should make this connection more clearly in the results.

Line 254 – A “lack of safety regulations in the United States” does not seem to apply to this research, as it occurred in S.A. Are the authors suggesting that melatonin should not be used in pediatrics ONLY in the United States due to a lack of safety regulations, or globally?  The statement also suggests that generally, there are a lack of safety regulations in the US.  This section should be clarified or removed.

Line 297 – “few pharmacist recommendations or physician prescriptions” – they account for almost 30% of the total, I would not consider this “few”. Suggest rewording.

General

The questions used in the study are not included in the submission, could this be added as a supplement?

Moderate edits required to improve clarity.

Author Response

Thanks very much

Reviewer 2 Report

The current manuscript is a quite interesting and well-done study on pharmacists general awareness of melatonin supplements in Jeddah, Saudi Arabia. Hence, I advise only some small changes before acceptance for publication:

- In the introduction section, a Figure representing the mechanism of action and the pharmacological and pharmacokinetic profiles of melatonin should be made and added;

- Further limitations of the current study should be addressed, such as the fact that it was conducted in one location only (Jeddah, Saudi Arabia) and for a short period of time (March–June 2023);

- The results of the current study should be better compared with other similar ones, performed in other countries, other continents.

Author Response

Thanks very much.
